# Improving Zero-shot Low-light Object Detection via Handling of Motion Blur

## Abstract

Zero-shot low-light object detection (ZLOD) presents great challenges as it aims to generalize detectors from the daytime domain to the nighttime domain without target data. Existing methods primarily focus on learning illumination consistency through daytime and synthetic nighttime image pairs, but they ignore a crucial characteristic that commonly coexists with low illumination, i.e., motion blur. Nighttime images are particularly susceptible to motion-induced blur due to the long exposure times of cameras. Thus, solely considering illumination reduction without motion blur may be sub-optimal for ZLOD. To this end, we propose a novel **I**llumination-**B**lur **C**onsistency (IBC) framework for ZLOD. Specifically, we synthesize nighttime images by considering illumination reduction and motion blur generation under a unified pipeline to access the complex nighttime domain. Then, we explore illumination-blur equivariant representations at the region and instance levels for better model adaptation. Consequently, IBC enables detectors to effectively generalize to the nighttime domain without relying on any dark data. Experimental results demonstrate the superior generalizability of our method. We also introduce a novel dataset named **NightVision** to expand the capacity of existing low-light benchmarks for community development.

## 1 Introduction

Low-light object detection is a challenging topic that aims to accurately localize objects in dark scenes characterized by low brightness and sharpness. However, advanced detectors Ren et al. (2016); Redmon & Farhadi (2018) are typically optimized on well-lit datasets (e.g., COCO Lin et al. (2014)), resulting in poor generalizability to nighttime scenarios. An intuitive scheme is to adopt image enhancement to enhance the visual quality of low-light images for benefiting detectors Guo et al. (2020); Cai et al. (2023); Feijoo et al. (2025). However, enhancement methods are biased toward human vision and ignore the significance of machine vision. Other research strives to enhance the performance of detectors by generating pseudo-labels or directly training on dark images Kennerley et al. (2023); Zhang et al. (2024); Wang et al. (2021; 2022). Nevertheless, these methods heavily rely on real-world nighttime data, which is scarce and hard to annotate.

To reduce reliance on nighttime data, some research introduces **zero-shot** day-night domain adaptation that aims to achieve adaptation without target data Lengyel et al. (2021); Luo et al. (2023). With this setting, this paper focuses on the detection task, extending it to zero-shot low-light object detection (ZLOD). Recent studies on this task primarily focus on consistency learning by daytime and synthetic nighttime image pairs. MAET Cui et al. (2021) explored the intrinsic pattern behind the illumination-degrading transformation. Sim-MinMax Luo et al. (2023) devised a similarity min-max paradigm to learn illumination-robust representations. DAI-Net Du et al. (2024) proposed to learn illumination invariance based on Retinex Theory Land (1977). However, these methods emphasize consistency learning only from the perspective of illumination reduction (as shown in Fig. 1), overlooking that motion blur is a crucial facet of real-world nighttime scenarios.

In low-light environments, the risk of motion blur significantly increases as the long exposure times required for capturing sharp photos can lead to unpredictable displacements of objects or cameras Zhou et al. (2022). The appearance of motion blur reflects a more realistic nighttime domain while posing greater challenges for detectors in dark scenes. As shown in Fig. 1, failed predictions from other methods indicate that the blurry features can greatly impair the performance of detec-

(a) Existing zero-shot methods    (b) Our method    (C) Predictions of a running *cat* in the dark by different methods

Figure 1: Motivation of this paper. In (a-b), existing zero-shot methods primarily focus on illumination consistency between daytime ($D$) and synthetic nighttime ($N$) images, while we explore illumination-blur consistency and achieve accurate predictions in (c).

tors in low-light conditions. Therefore, further investigating the impact of motion blur under low illumination conditions may be an effective solution to improve ZLOD.

To this end, we propose a novel Illumination-Blur Consistency (IBC) framework for ZLOD. Specifically, we establish a connection between illumination reduction and motion blur generation through a controllable exposure factor, coming up with an exposure-guided nighttime pipeline (ENP). This pipeline synthesizes realistic nighttime images while enabling models to learn the coexistence of low illumination and motion blur. Notably, the generation of motion blur on synthetic images changes the spatial position of objects relative to the original. Hence, we fully consider this feature misalignment between daytime images and their synthetic counterparts and innovatively propose a multi-level model adaptation (MMA) to encourage region-wise and instance-wise feature consistency, enabling detectors to learn spatial and global illumination-blur equivariant representations for better model adaptation. Consequently, our IBC effectively achieves domain adaptation without relying on any real-world nighttime data. Experimental results on low-light datasets highlight the superior generalizability of our method in nighttime scenarios. In addition to IBC, we also build a novel dataset named NightVision to alleviate the demand for dark data in the community. NightVision includes 10.0k low-light images and 56.0k objects of 18 categories, expanding the capacity of existing low-light data with great diversity. Our contributions can be summarized as follows:

- We reveal that the neglect of motion blur limits the generalizability of existing methods in dark scenes and provide an insight of further investigating motion blur under low illumination conditions to improve ZLOD.

- We propose a novel Illumination-Blur Consistency (IBC) framework to learn illumination-blur equivariant representations, enhancing the low-light generalizability of detectors without relying on any nighttime data.

- Experimental results on low-light datasets highlight the superior generalizability of our method in nighttime scenarios compared to other competitive methods.

- We build a novel dataset named NightVision to facilitate community research. To our knowledge, NightVision contains the largest amounts of categories for low-light object detection with great generality.

## 2   RELATED WORKS

**Motion Blur Learning.** Motion blur, prevalent in dynamic scenes, is widely studied across vision tasks. Image deblurring is a typical low-level task that improves the clarity of blurry images Bahat et al. (2017); Zamir et al. (2022). In high-level vision, motion blur is treated as a specific disturbance, with studies investigating its impact on tasks like classification Vasiljevic et al. (2016), detection Sayed & Brostow (2021) and segmentation Rajagopalan et al. (2023).

**Low-light Object Detection.** Despite established benchmarks for dark scene perception Neumann et al. (2019); Loh & Chan (2019); Yang et al. (2020); Morawski et al. (2022); Yu et al. (2020); Hong et al. (2021); Chen et al. (2023), low-light object detection remains challenging. An intuitive scheme is to brighten low-light images for benefiting detectors Zhang et al. (2019); Guo et al. (2020); Cai et al. (2023), and some methods further integrated low-light image enhancement and deblurring by exploiting their interrelated nature Zhou et al. (2022); Lv et al. (2024); Feijoo et al. (2025). Other solutions involve arousing the potential of image restoration for detectors Hashmi et al. (2023); Cui et al. (2024) or improving detection performance from the perspective of RAW data Chen et al.

Figure 2: Examples of NightVision.

Table 1: Comparison of Low-Light Datasets

| Dataset | #Images | #Boxes | #Category | Multi-angle | Multi-category | Multi-weather |
|---------|---------|--------|-----------|-------------|----------------|---------------|
| Darkface | 6.0k | 50.4k | 1 | ✓ | | |
| NightOwls | 36.7k | 61.0k | 1 | | | ✓ |
| RAW-NOD | 7.2k | 46.8k | 3 | ✓ | ✓ | |
| LIS | 2.2k | 10.5k | 8 | ✓ | ✓ | |
| BDD100K | 31.9k | 509.6k | 10 | | ✓ | ✓ |
| ExDark | 7.4k | 23.7k | 12 | ✓ | ✓ | ✓ |
| NightVision | 10.0k | 56.0k | 18 | ✓ | ✓ | ✓ |

(2023); Guo et al. (2025). Another research area focuses on generating pseudo-labels or directly training on dark images Kennerley et al. (2023); Zhang et al. (2024); Wang et al. (2021; 2022), which greatly relies on real-world nighttime data. To reduce the reliance on dark data, **zero-shot** day-night domain adaptation is proposed as a more challenging setting that requires models to generalize to the nighttime domain without dark data Lengyel et al. (2021); Luo et al. (2023). With this setting, CIConv Lengyel et al. (2021) proposed a physics prior by a color invariant convolution layer to address domain shift. MAET Cui et al. (2021) explored the intrinsic representation of illumination degradation. Sim-MinMax Luo et al. (2023) devised a similarity min-max paradigm to learn illumination-robust representations for zero-shot learning. DAI-Net Du et al. (2024) focused on learning illumination invariance for object detection based on Retinex Theory.

Despite these advances, image enhancement faces human bias and real-time limitations, while prior zero-shot methods neglect the impact of motion blur on nighttime perception. Differently, we deeply analyze motion blur in dark scenes and devise a novel framework to enhance ZLOD.

## 3 NIGHTVISION DATASET

**Data Construction.** Large-scale, high-quality datasets are critical for domain exploration, yet existing low-light object detection datasets have limited data capacity (e.g., categories, scenarios), hindering a thorough understanding of this task. Hence, we collect a novel dataset named NightVision for future research. During data collection, we ensure generality in three key aspects: 1) All images are from various devices to avoid bias from a single shooting system. 2) Diverse geographical/weather conditions are fully considered for real-world fidelity. 3) Both indoor and outdoor scenes contain objects of varying sizes to enhance applicability. Ultimately, we select 10.0k web-sourced sRGB images to form NightVision without overlapping with existing public datasets. After image collection, we annotated 56.0k bounding boxes across 18 categories using X-AnyLabeling Wang (2023). To minimize errors, we carefully examine all annotation results to confirm the correct category and bounding box assignments. Some examples can be seen in Fig. 2. As for data split, our NightVision is split (5:2:3 ratio) into training (4,997), validation (1,989), and test (3,037) sets. Moreover, the assignment of images from different scenes is fully considered to prevent distribution shifts.

**Data Statistics and Analysis.** We compare NightVision with DarkFace Yang et al. (2020), NightOwls Neumann et al. (2019), RAW-NOD Morawski et al. (2022), LIS Chen et al. (2023), the nighttime part of BDD100K Yu et al. (2020), and ExDark Loh & Chan (2019) datasets in Table 1. Though NightOwls and BDD100K contain large nighttime data, they mainly focus on traffic conditions at parallel angles. While Darkface, RAW-NOD and LIS encompass multiple camera angles, their limited categories and weather conditions constrain their generality. Compared to Ex-Dark, the widely used benchmark for low-light object detection, NightVision offers greater data capacity, wider object size variance, and broader illumination distribution, underscoring its higher complexity (see Appendix A for details). To our knowledge, our NightVision has the largest number of categories for low-light object detection, and we hope it will advance the community.

## 4 ILLUMINATION-BLUR CONSISTENCY

### 4.1 OVERVIEW

Existing ZLOD methods primarily address day-night representation disparity by learning illumination consistency though a designed pixel-wise illumination-reduced translation, yet neglect the impact of motion blur on nighttime perception, limiting low-light generalizability.

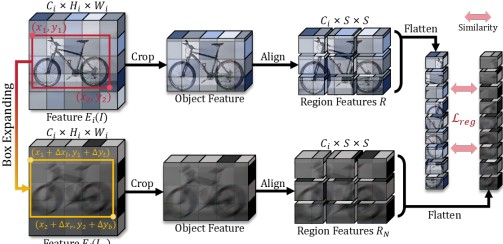

Figure 3: Overall of Illumination-Blur Consistency (IBC).

Figure 4: An illustration of region-level consistency.

Moving beyond illumination consistency, we propose a novel Illumination-Blur Consistency (IBC) framework (Fig. 3) to enhance ZLOD, which consists of an exposure-guided nighttime pipeline (ENP) and multi-level model adaptation (MMA). The ENP synthesizes domain-realistic nighttime images by progressively darkening and blurring daytime images under the guidance of exposure, while the MMA drives consistency learning between the daytime and synthetic nighttime images at region and instance levels for better adaptation.

## 4.2 EXPOSURE-GUIDED NIGHTTIME PIPELINE

The ENP unifies illumination reduction and motion blur synthesis, explicitly modeling their joint occurrence in low-light environments. Notably, we employ a single exposure factor to concurrently regulate global brightness (Image Darkening) and motion blur intensity (Image Blurring), which better approximates physical low-light image formation.

### 4.2.1 IMAGE DARKENING

Different from GAN-based Zhu et al. (2017); Karras et al. (2019) or ISP-based Cui et al. (2021) methods, where the former relies on real dark data and the latter is biased toward specific camera sensors, we develop a darkening function with learnable parameters based on Luo et al. (2023). This function adjusts the image brightness by mapping pixel values to the expected exposure. Next, we detail the mapping process and darkening function.

**Mapping Process.** The mapping function $F$ is designed to nonlinearly adjust the pixel intensity $x_i \in [0, 1]$ to an expected value $y_i \in [0, 1]$. To avoid information loss caused by overflow, $F$ should satisfy the boundary conditions:

$$F(x_i) = \begin{cases} 0 & \text{if } x_i = 0 \\ 1 & \text{if } x_i = 1. \end{cases} \tag{1}$$

Following Guo et al. (2020), we adopt the iterative curve $F(x_i) = f^8(x_i)$, where $f(x_i) = \alpha \cdot x_i^2 + (1 - \alpha) \cdot x_i$ and $\alpha \in [-1, 1]$ is a learnable parameter. Therefore, the mapping function is defined as $y_i = F(x_i, \alpha)$. To illustrate this at the image level, given a normalized image $I \in [0, 1]^{C \times H \times W}$ and a learnable adjustment map $A \in [-1, 1]^{H \times W}$, the mapping process can be formulated as $I_m = F(I, A)$, where $I_m$ is the mapping output.

**Darkening Function.** To achieve image darkening, we adopt a network to estimate the appropriate adjustment map $A$ for illumination reduction. Specifically, we train an estimation network to produce the corresponding $A$ for the mapping process according to the input controllable exposure $e$ and well-lit image $I$. This allows us to simulate different illumination conditions of the image $I$ by the controllable exposure $e$. However, due to the constraints in Eq. 1, the pixel $x_i = 1$ could not be adjusted by any $A$. To address this limitation, we divide all the pixel values by an operation $d(x_i, \beta) = x_i / \beta$ before performing mapping, and finally rescale the mapping output back to the original space. Notably, $d$ is parameterized by another learnable map $B$. Consequently, we can synthesize realistic low-light image $I_D$ and avoid violating the constraints of the mapping function $F$. The entire process can be formulated as:

$$\mathcal{D}(I, e) = d^{-1}(F(d(I, B), A), B), \tag{2}$$

where $A$ and $B$ are both estimated by the estimation network and $\mathcal{D}(,)$ is the darkening function. Notably, $\mathcal{D}(I, e)$ is equivalent to $I_D$. The details of estimation network training and noise injection are provided in Appendix B and Appendix C, respectively.

### 4.2.2 IMAGE BLURRING

Motion blur is common in nighttime photography since longer exposure times of cameras increase the risk of egomotion-induced blur in low-light conditions, which is ignored by existing ZLOD works. In contrast, we focus on the coexistence relationship between low illumination and motion blur and further generate motion blur for $I_D$ based on its exposure $e$ in Eq. 2 to simulate more realistic low-light environments. Inspired by Sayed & Brostow (2021), we generates blur kernels to achieve image blurring.

**Blurring Function.** Each blur kernel is generated based on a 2D trajectory and an exposure time. To create each point of the trajectory $x = (x_0, \ldots, x_t, \ldots, x_l)$, we first simulate three types of camera motion $M_i \in \{M_1, M_2, M_3\}$ by fixed parameters, where $M_1$ represents a nervous camera state, $M_2$ denotes back-and-forth motion, and $M_3$ signifies straight rectilinear movement. Then we initialize a camera state $(x_0, v_0)$ in the complex space, where $x_0$ is the original position and $v_0$ is a velocity vector randomly sampled from the unit circle. At each iteration $t$, the velocity vector $v_t$ of the camera is updated based on the previous state and a random camera jerk:

$$v_t = M_i \cdot (v_n - Tx_{t-1}) + \underbrace{2M_i \cdot |v_{t-1}| \cdot v_j}_{\text{camera jerk}}, \tag{3}$$

where $v_n$ and $v_j$ are random velocity vectors sample from $\mathcal{N}(0, 1)$ and the unit circle, respectively, $Tx_{t-1}$ is an inertial tendency at the position $x_{t-1}$, and $M_i$ represents the camera motion defined above. Then, the position $x_t$ can be computed based on the previous position $x_{t-1}$ as follows:

$$x_t = x_{t-1} + v_t \cdot (l/L), \tag{4}$$

where $l$ is the trajectory length (defaulted to 96), and $L$ is the total number of iterations.

Moreover, to simulate the effect of exposure time on trajectory, we truncate the trajectory length $l$ based on the controllable exposure factor $e$:

$$x = (x_0, \ldots, x_{\lfloor l \times e \rfloor - 1}). \tag{5}$$

Next, we initialize a 2D zero matrix with size $k$, then map each point $x_i$ of trajectory $x$ to obtain the blur kernel $K$:

$$K[\lfloor Re(x_i) \rfloor, \lfloor Im(x_i) \rfloor] = 1, \tag{6}$$

where $Re(x_i)$ and $Im(x_i)$ represent real quantity and imaginary quantity of the trajectory point $x_i$, respectively. Consequently, given the synthetic low-light image $I_D$, the process of image blurring can be formulated as:

$$\mathcal{B}(I_D, e) = I_D^c * K, c \in \{R, G, B\}, \tag{7}$$

where $c$ denotes the color channel, $\mathcal{B}$ represents the blurring function, and blur kernel $K$ is generated from a random trajectory $x$ and the controllable exposure factor $e$. Similarly, $\mathcal{B}(I_D, e)$ is equivalent to $I_N$.

**Box Expanding.** After the blurring as described in Eq. 7, the boundaries of each object in the image are changed. To avoid information loss, we expand each original box to fit the boundaries of the blurred objects. Defining an original box by two coordinates $(x_1, y_1, x_2, y_2)$, the box expanding process can be formulated as:

$$\begin{aligned} x_1' &= x_1 + \Delta x_l, \quad y_1' = y_1 + \Delta y_t, \\ x_2' &= x_2 + \Delta x_r, \quad y_2' = y_2 + \Delta y_b, \end{aligned} \tag{8}$$

where $\Delta x_l$, $\Delta x_r$, $\Delta y_t$, and $\Delta y_b$ are box expansions in the left, right, top, and bottom, respectively.

Overall, the ENP generates a nighttime image $I_N$ by gradually darkening and blurring the input $I$ under the control of exposure $e$, enabling exploration of a realistic nighttime domain (see Appendix D for comparisons of different low-light synthesis methods). Both $I$ and $I_N$ are then fed into the following multi-level model adaptation for consistency learning.

### 4.3 MULTI-LEVEL MODEL ADAPTATION

Since the generation of motion blur changes the spatial position of each blurred object feature relative to the original, a simple vector-based (i.e., instance-wise) alignment Luo et al. (2023) is not robust enough to capture dense consistency representations for object detection. To this end, we further introduce region-wise consistency to adaptively align features between original and blurred objects, addressing motion blur-induced spatial misalignment (validated in Appendix E).

Hence, given an input $I$ and its synthetic image $I_N$, their last three layer features $E(I)$ and $E(I_N)$ extracted by a shared encoder $E$ are encouraged consistency at region and instance levels, resulting in multi-level model adaptation.

#### 4.3.1 REGION-LEVEL CONSISTENCY

As shown in Fig. 4, given the $i$-th layer feature $E_i(I) \in \mathbb{R}^{C_i \times H_i \times W_i}$ and its bounding box of the object ($i$-th layer depends on the size of the bounding box according to the assignment strategy of Lin et al. (2017)). We first accurately crop the object feature from $E_i(I)$ based on the bounding box. Next, we align the object feature to $S \times S$ region features, where each region feature is computed by bilinear interpolation using its surrounding feature points in $E_i(I)$ inspired by He et al. (2017). This ensures that object features of different sizes can be scaled to a uniform size for consistency learning.

By applying the Crop and Align processes to both $E_i(I)$ and $E_i(I_N)$, we can obtain two region features, denoted as $R \in \mathbb{R}^{C_i \times S \times S}$ and $R_N \in \mathbb{R}^{C_i \times S \times S}$. These features are then flattened to maximize the similarity of each pair, enabling the computation of the consistency loss $\mathcal{L}_{reg}$:

$$\mathcal{L}_{reg} = -\frac{\sum_i \mathcal{C}(s(p_r(r^i)), z_r(r^i_N)) + \mathcal{C}(z_r(r^i), s(p_r(r^i_N)))}{2}, \tag{9}$$

where $r^i \in R$ and $r^i_N \in R_N$ denote a region feature pair, $p_r$ and $z_r$ are projection head and prediction head Xie et al. (2021) at the region level, respectively, $s$ is the operation of stop-gradient to prevent model collapse Chen & He (2021), and $\mathcal{C}(,)$ is a cosine function. The loss $\mathcal{L}_{reg}$ is averaged over all region feature pairs and further averaged in a batch to drive representation learning.

#### 4.3.2 INSTANCE-LEVEL CONSISTENCY

In addition to region-level consistency, we also introduce instance-level consistency to learn more robust representations from global information. Specifically, we pool and flatten the last layer features of $E(I)$ and $E(I_N)$, resulting in two feature vectors $v$ and $v_N$, respectively. Then we maximize their similarity by a consistency loss $\mathcal{L}_{ins}$:

$$\mathcal{L}_{ins} = -\frac{\mathcal{C}(s(p_i(v)), z_i(v_N)) + \mathcal{C}(z_i(v), s(p_i(v_N)))}{2}, \tag{10}$$

where $p_i$ and $z_i$ are projection head and prediction head (initialized by MLPs Xie et al. (2021)) at the instance level, respectively.

Besides the multi-level consistency loss $\mathcal{L}_{con}$, we also incorporate supervision detection loss $\mathcal{L}_{Det}$ to the final loss:

$$\mathcal{L}_{con} = \mathcal{L}_{reg} + \alpha \cdot \mathcal{L}_{ins}, \quad \mathcal{L} = \mathcal{L}_{Det} + \beta \cdot \mathcal{L}_{con}. \tag{11}$$

where $\alpha$ and $\beta$ are balanced factors, both defaulting to 1.

The proposed consistency losses $\mathcal{L}_{reg}$ and $\mathcal{L}_{ins}$ are mutually beneficial: $\mathcal{L}_{reg}$ learns spatial consistency representations and $\mathcal{L}_{ins}$ explores the global consistency information, enabling detectors to learn multi-level illumination-blur equivariant representations for better model adaptation. In addition, the incorporation of supervision detection loss $\mathcal{L}_{Det}$ contributes to driving discriminative representation learning for low-light object detection.

### 4.4 MODEL TRAINING AND INFERENCE

**Training.** We first train the estimation network of ENP to generate appropriate adjustment maps $A$ and $B$ through the controllable exposure $e$ that sampled uniformly in [0, 0.5]. During IBC training, the estimation network remains frozen to produce illumination-reduced examples guided by

Table 2: Comparison results on NightVision and ExDark Loh & Chan (2019). + denotes enhancement method.

| Methods | NightVision Recall | NightVision mAP | ExDark Recall | ExDark mAP |
|---|---|---|---|---|
| *Zero-shot* | | | | |
| YOLOv3 | 0.629 | 0.474 | 0.673 | 0.503 |
| +Retinexformer | 0.606 | 0.455 | 0.668 | 0.490 |
| +Zero-DCE | 0.615 | 0.463 | 0.672 | 0.500 |
| +LEDNet | 0.629 | 0.472 | 0.680 | 0.510 |
| +FourierDiff | 0.644 | 0.483 | 0.665 | 0.490 |
| +DarkIR | 0.639 | 0.477 | 0.673 | 0.504 |
| MAET | 0.646 | 0.486 | 0.683 | 0.512 |
| CIConv | 0.657 | 0.495 | 0.690 | 0.518 |
| Sim-MinMax | 0.660 | 0.500 | 0.693 | 0.520 |
| DAI-Net | 0.665 | 0.505 | 0.698 | 0.522 |
| **IBC (Ours)** | **0.683** | **0.525** | **0.714** | **0.539** |
| *Fine-tuned* | | | | |
| YOLOv3 | 0.734 | 0.604 | 0.832 | 0.678 |
| +Retinexformer | 0.704 | 0.577 | 0.815 | 0.664 |
| +Zero-DCE | 0.710 | 0.584 | 0.819 | 0.668 |
| +LEDNet | 0.709 | 0.580 | 0.820 | 0.670 |
| +FourierDiff | 0.715 | 0.594 | 0.815 | 0.666 |
| +DarkIR | 0.719 | 0.595 | 0.820 | 0.669 |
| MAET | 0.739 | 0.609 | 0.836 | 0.684 |
| CIConv | 0.743 | 0.614 | 0.843 | 0.692 |
| IA-YOLO | 0.740 | 0.610 | 0.834 | 0.683 |
| Featenhancer | 0.736 | 0.609 | 0.837 | 0.685 |
| Sim-MinMax | 0.745 | 0.617 | 0.847 | 0.696 |
| YOLA | 0.745 | 0.621 | 0.845 | 0.695 |
| DAI-Net | 0.747 | 0.620 | 0.852 | 0.699 |
| **IBC (Ours)** | **0.759** | **0.635** | **0.868** | **0.713** |

Table 3: Comparison results (mAP) of Sharp, Synthetic blurry (Syn.), and Real-world blurry (Real.) evaluations.

| Methods | COVO$_{18}\rightarrow$NightVision Sharp | Syn. | Real. | COCO$_{12}\rightarrow$ExDark Sharp | Syn. | Real. |
|---|---|---|---|---|---|---|
| YOLOv3 | 0.480 | 0.401 | 0.469 | 0.552 | 0.427 | 0.483 |
| +Retinexformer | 0.468 | 0.388 | 0.444 | 0.548 | 0.416 | 0.465 |
| +Zero-DCE | 0.478 | 0.395 | 0.449 | 0.551 | 0.421 | 0.472 |
| +LEDNet | 0.466 | 0.385 | 0.435 | 0.535 | 0.402 | 0.460 |
| +FourierDiff | 0.470 | 0.392 | 0.440 | 0.546 | 0.414 | 0.455 |
| +DarkIR | 0.479 | 0.393 | 0.433 | 0.550 | 0.415 | 0.460 |
| MAET | 0.501 | 0.416 | 0.470 | 0.578 | 0.445 | 0.492 |
| CIConv | 0.516 | 0.419 | 0.472 | 0.589 | 0.450 | 0.501 |
| Sim-MinMax | 0.520 | 0.426 | 0.479 | 0.594 | 0.453 | 0.505 |
| DAI-Net | 0.525 | 0.428 | 0.486 | **0.602** | 0.451 | 0.512 |
| **IBC (Ours)** | **0.535** | **0.467** | **0.521** | 0.600 | **0.492** | **0.532** |

Table 4: Ablation results (mAP) on NightVision and ExDark Loh & Chan (2019). $\mathcal{D}$ and $\mathcal{B}$ are Image Darkening and Image Blurring, respectively.

| Methods | ENP $\mathcal{D}$ | ENP $\mathcal{B}$ | MMA $\mathcal{L}_{reg}$ | MMA $\mathcal{L}_{ins}$ | NightVision | ExDark |
|---|---|---|---|---|---|---|
| Baseline | | | | | 0.474 | 0.503 |
| IBC-A | ✓ | | | | 0.496 | 0.517 |
| IBC-B | | ✓ | | | 0.478 | 0.506 |
| IBC-C | ✓ | ✓ | | | 0.504 | 0.520 |
| IBC-D | ✓ | ✓ | ✓ | | 0.519 | 0.534 |
| IBC-E | ✓ | ✓ | | ✓ | 0.512 | 0.526 |
| IBC-F | ✓ | | ✓ | ✓ | 0.503 | 0.520 |
| **IBC (Ours)** | ✓ | ✓ | ✓ | ✓ | **0.525** | **0.539** |

different $e$. These examples are then fed with corresponding exposures into image blurring (with probability $\rho = 0.75$) to generate the final synthetic nighttime images. Unlike training the estimation network, $e$ is uniformly sampled in [0, 0.2] to simulate different poor illumination conditions. In the MMA, the encoder $E$ serves as a feature extractor, and the detector head is exploited to decode the learnt features from $E$.

**Inference.** During testing, images are only fed into the encoder and detector head to predict categories and locations of objects. This will not introduce additional parameters and inference time to the vanilla detector.

## 5 EXPERIMENTS

### 5.1 LOW-LIGHT OBJECT DETECTION

**Settings.** We compare IBC with Zero-DCE Guo et al. (2020), DarkIR Feijoo et al. (2025), Retinexformer Cai et al. (2023), LEDNet Zhou et al. (2022), FourierDiff Lv et al. (2024), MAET Cui et al. (2021), CIConv Lengyel et al. (2021), IA-YOLO Liu et al. (2022), Featenhancer Hashmi et al. (2023), Sim-MinMax Luo et al. (2023), YOLA Hong et al. (2024) and DAI-Net Du et al. (2024) on the NightVision and ExDark Loh & Chan (2019) datasets. Notably, enhancement methods are adopted as preprocessing steps.

To meet the zero-shot setting, we sample daytime data from COCO Lin et al. (2014) and PASCAL VOC Everingham et al. (2010) based on the categories of nighttime datasets, constructing two dataset pairs: COVO$_{18}$−NightVision and COCO$_{12}$−ExDark. Following Cui et al. (2021); Du et al. (2024), we adopt YOLOv3 Redmon & Farhadi (2018) as the vanilla detector for all methods. The training of the estimation network is consistent in Luo et al. (2023). Images are resized to $416 \times 416$ before being fed into the detectors. We measure detection performance via mean Average Precision (mAP) and recall at the IoU threshold of 0.5. More information about daytime datasets and implementation details are provided in Appendix F.

**Zero-shot Evaluation.** In this part, we train all zero-shot methods on the training set of daytime datasets and evaluate them on the test set of nighttime datasets. The experimental results presented in Table 2 show that enhancement methods attempt to facilitate human vision but fail to handle the complex low-light conditions for object detection, resulting in limited performance. Although other zero-shot methods improve the performance of the vanilla YOLOv3 to varying degrees, the performance gains are limited due to their insufficient consideration of the characteristics of dark images. In contrast, our method yields superior results in all comparison methods and enhances the baseline by 5.4% mAP (NightVision) and 3.6% mAP (ExDark), demonstrating the generalizability of our method which considers motion blur for ZLOD. In addition, we also perform zero-shot evaluations on RAW-NOD Morawski et al. (2022) and BDD100K Yu et al. (2020) datasets in Appendix G, where the results further verify the effectiveness of our IBC.

**Fine-tuned Evaluation.** In this part, we further fine-tune and evaluate all the methods on the nighttime dataset. The results in Table 2 show that our method generally achieves the best results, demonstrating that further considering motion blur contributes to searching for a more optimal pre-trained feature space of the nighttime domain.

## 5.2 DETECTION WITH BLURRY DATA

This section analyzes the impact of blurry data on low-light object detection. To acquire real-world blurry data, we select images with varying degrees of blur from the nighttime dataset (3,551 images from NightVision and 2,748 images from ExDark), leaving the rest as sharp data.

**Synthetic Blurry Evaluation.** We first synthesize blurry images from sharp images based on Sayed & Brostow (2021) and then evaluate the zero-shot methods on sharp images and synthetic blurry images, respectively.

The results in Table 3 show that all methods exhibit significant performance degradation with synthetic blurry images compared to sharp images. For instance, on the ExDark dataset, DAI-Net Du et al. (2024) effectively learns reflectance and illumination with sharp images but struggles with synthetic blurry scenes (60.2% → 45.1% mAP). By further considering motion blur, our method presents less performance degradation and outperforms other methods in synthetic blurry scenes.

**Real-world Blurry Evaluation.** We also evaluate the zero-shot methods on the real-world blurry data of nighttime datasets. The results in Table 3 show that our method remains superior among the comparison methods, further validating its robustness against low illumination and motion blur in real-world nighttime scenarios.

## 5.3 ABLATION STUDY

Here we conduct ablation studies to justify ENP and MMA, as shown in Table 4.

As for IBC-A, we train the baseline with synthetic images darkened by Image Darkening. The results show a great average improvement of 1.8% mAP, indicating its effectiveness in simulating low illumination conditions.

For IBC-B, we adopt daytime images blurred by Image Blurring for training, slightly boosting the baseline by an average of 0.35% mAP. This implies that simply applying motion blur to well-lit images is not robust enough to address the large gap between daytime and nighttime domains.

By combining Image Darkening and Image Blurring in a unified pipeline, IBC-C enhances the baseline by an average of 2.35% mAP, verifying the strong capability of ENP in simulating nighttime scenarios.

The region-level consistency loss $\mathcal{L}_{reg}$ seeks to encourage spatial consistency between each region feature pair. Solely considering this loss, IBC-D presents an average improvement of 1.45% mAP, indicating its ability to learn robust spatial representation for illumination-blur consistency.

The instance-level consistency loss $\mathcal{L}_{ins}$ tries to align the instance features for learning global consistency information. With this loss, IBC-E brings an average enhancement of 0.7% mAP. Compared to $\mathcal{L}_{reg}$, this lower result is probably because the instance-level consistency contributes less to learning spatial representation in dense vision tasks.

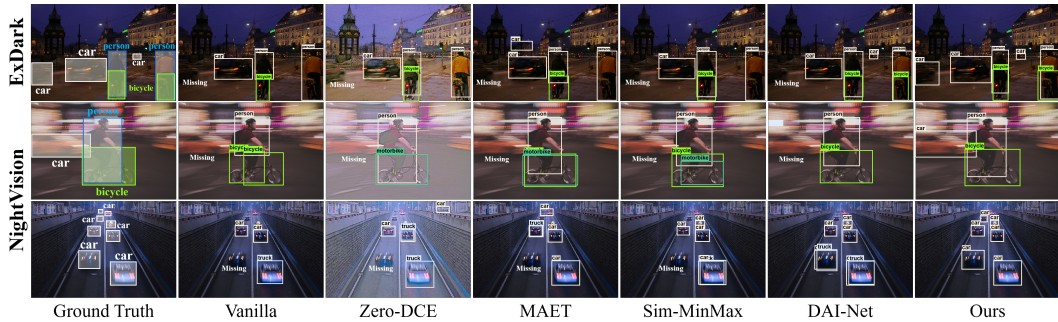

Figure 5: Prediction visualizations by different methods. The first row is from ExDark, and the rest comes from NightVision.

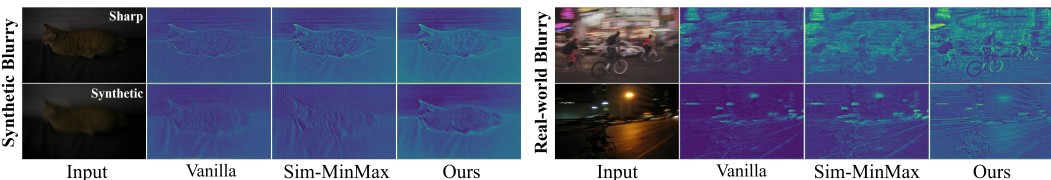

Figure 6: Backbone feature visualizations by different methods on synthesis (left) and real-world (right) blurry evaluation.

Without Image Blurring in the ENP, IBC-F greatly degrade IBC's performance by 2.05% mAP, verifying the importance of considering motion blur in low light conditions.

Combined with ENP and MMA, our IBC improves the average performance of baseline by 4.35% mAP and IBC-C by 2.0% mAP, proving the effectiveness of MMA in learning illumination-blur equivariant representations for ZLOD.

Please see Appendix H for more ablation studies of hyper-parameters and box expanding.

### 5.4 VISUALIZATION

We visualize the prediction results (real-world in Fig. 5 and synthetic in Fig. 12 of Appendix I) and backbone features (Fig. 6) to highlight the strong generalizability of our method.

In Fig. 5, the results from NightVision show that other methods misidentify blurry cars as trucks and miss blurry or small-sized objects, while our method accurately detects most objects in dark dynamic scenes. These results also highlight the challenges of NightVision, which requires stronger fine-grained and small-sized detection capabilities than ExDark. In Fig. 6, synthetic blurry features from the vanilla model and Sim-MinMax Du et al. (2024) illustrate how motion blur degrades a detector from the feature perspective, while our method remains robust and captures more detailed object representations in both synthetic and real-world blurry nighttime scenarios.

### 6 CONCLUSION

This paper studies zero-shot low-light object detection (ZLOD) that aims to generalize detectors from the daytime domain to the nighttime domain without relying on target data. We reveal the neglect of motion blur in existing methods and propose an effective Illumination-Blur Consistency (IBC) framework to explore the multi-level illumination-blur equivariant representations for improving ZLOD. Experimental results demonstrate the superior low-light generalizability of our method. Moreover, we build a considerable dataset named NightVision to expand the capacity of existing low-light benchmarks with great diversity. We hope this paper can inspire more insightful research and facilitate further development in the community.

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

APPENDIX

Our project is at `https://anonymous.4open.science/r/nv1.`. In this appendix, we include the following additional discussions:

- Appendix A provides more detailed statistics of NightVision.
- Appendix B introduces the training objectives of estimation network.
- Appendix C describes the noise injection during the process of synthesizing low-light images.
- Appendix D compares the performance of different low-light synthesis methods.
- Appendix E explores the motion blur-induced feature misalignment.
- Appendix F describes more implementation details, including the training settings and daytime datasets.
- Appendix G provides additional zero-shot evaluations on BDD100K Yu et al. (2020) and RAW-NOD Morawski et al. (2022) datasets.
- Appendix H presents more ablation studies, including hyper-parameters and box expanding.
- Appendix I includes the visualization of synthetic blurry evaluation.
- Appendix J explores the generalization of our method to other domains, including the RAW nighttime domain and daytime domain.
- Appendix K describes the detailed ethics statement.
- Appendix L describes the usage of the large language model.

## A DETAILED STATISTICS OF NIGHTVISION

### A.1 COMPARISON WITH EXDARK

To highlight the challenge of NightVision, we present a detailed statistical comparison with ExDark Loh & Chan (2019):

**More categories and objects.** As illustrated in Fig. 7a, compared to ExDark, our NightVision generally includes more objects per category and six additional categories, including *umbrella*, *bench*, *truck*, *aeroplane*, *bed* and *sofa*. With the additional objects available for each category and the broader spectrum of categories, researchers enable a deeper exploration of intra-class and inter-class variability, allowing models to learn nuanced features and improve their generalization capabilities.

**Wider size variance of objects.** As shown in Fig. 7b, our NightVision covers a broader range of object sizes than ExDark, containing not only larger objects but also smaller objects that are harder to recognize in challenging low-light conditions. This substantial variance in object sizes is more in line with real-world nighttime scenarios.

**Broader illumination distribution.** We quantify the illumination distribution of each dataset by calculating the average pixel intensity per image. As illustrated in Fig. 7c, NightVision has a broader low illumination distribution (30–80) than ExDark's, greatly expanding nighttime illumination diversity. This broader spectrum aligns more closely with the varied illumination conditions encountered in real-world dark environments.

### A.2 STATISTICS OF MOTION BLUR

To investigate motion blur in nighttime images, we analyze the frequency of motion blur within the NightVision dataset. As shown in Fig. 8, our statistics reveal that 35.4% of NightVision images contain noticeable motion blur (while only **3.2%** in daytime COCO exhibit this issue). This discrepancy is particularly pronounced in dynamic scenarios like traffic (e.g., moving vehicles) compared to static indoor settings (e.g., stationary furniture). Such a significant gap underscores the importance of addressing motion blur for robust low-light object detection, especially in dynamic environments. These findings motivate our exploration of illumination-blur consistency for zero-shot low-light

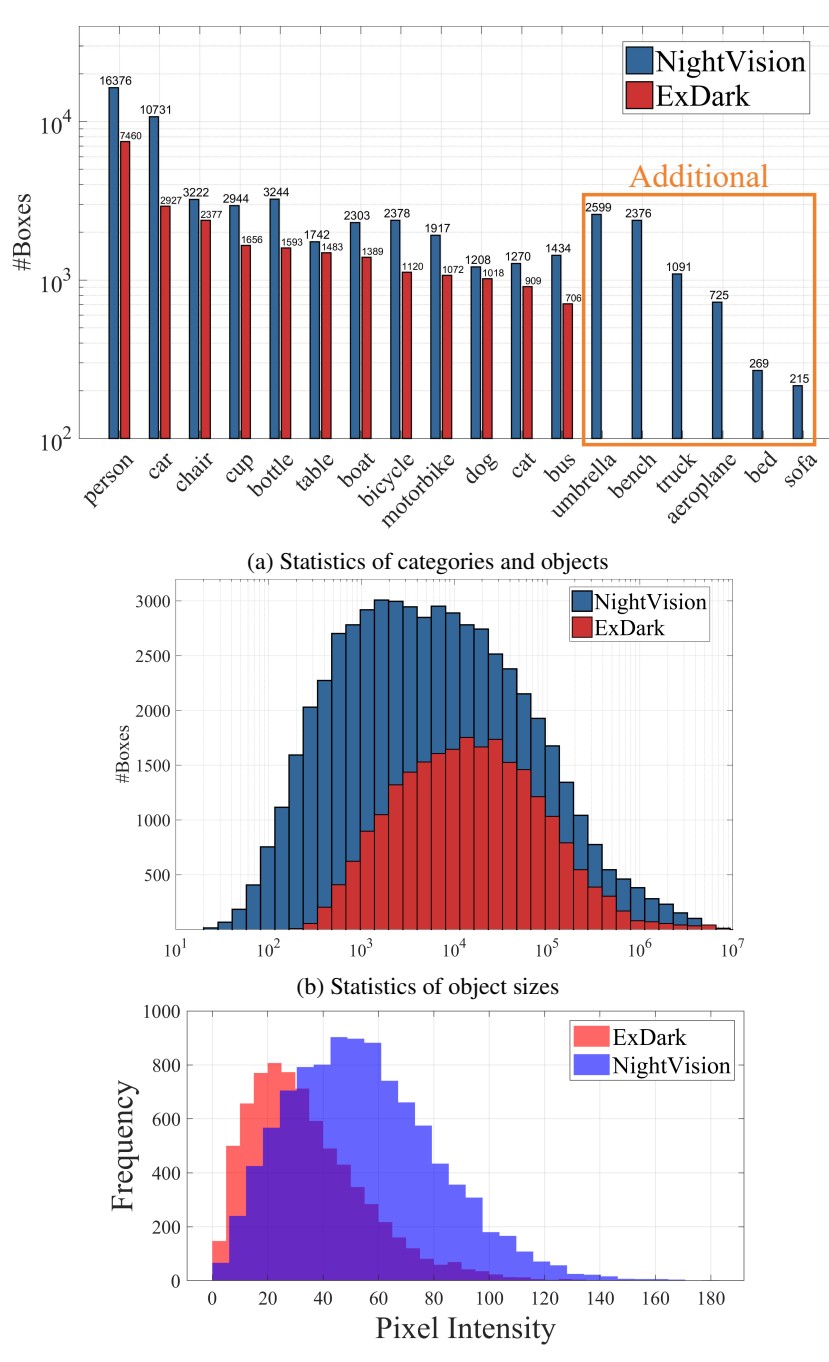

(a) Statistics of categories and objects

(b) Statistics of object sizes

(c) Statistics of illumination distribution

Figure 7: Statistical information of NightVision and ExDark Loh & Chan (2019).

object detection, as we found that most previous methods primarily focus solely on illumination consistency.

## B    ESTIMATION NETWORK TRAINING

In the darkening process, the learnable maps $A$ and $B$ are estimated by an estimation network. In this paper, we adopt a U-Net Ronneberger et al. (2015) as the estimation network and four training losses Luo et al. (2023) as regularization.

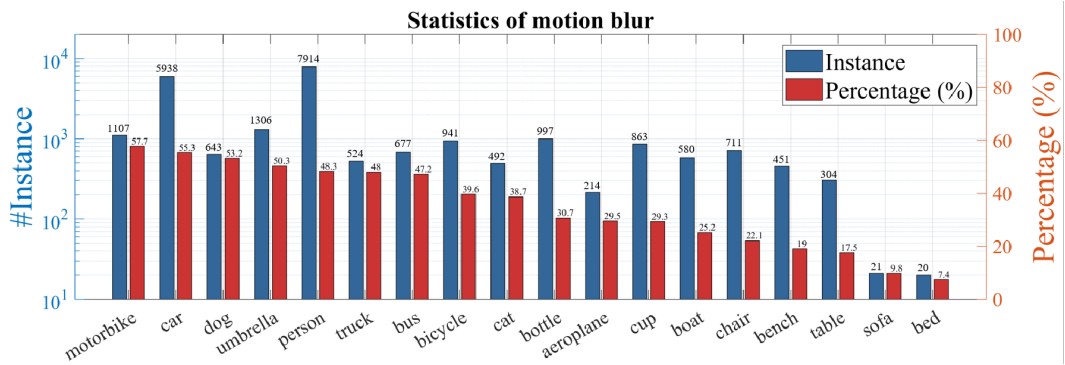

Figure 8: Statistics of motion blur in NightVision. The number and percentage of blurry images for each category are provided.

Following Guo et al. (2020), a color constancy loss $\mathcal{L}_C$ is introduced to correct color deviations:

$$\mathcal{L}_C = \sum_{\forall (m,n) \in \varepsilon} (J^m - J^n)^2, \varepsilon = \{(R,G),(R,B),(G,B)\} \tag{12}$$

To control the exposure of the synthesized image, the input exposure $e$ is aligned with the channel average of $I_D$:

$$\mathcal{L}_E = \frac{1}{M} \sum_i^M |\bar{x}_i - e|, \tag{13}$$

where $\bar{x}_i$ represents the channel-wise average value of pixel $x_i$ and $M$ is the total number of pixels.

The exposure adjustment map $A$ is adopted to reduce the illumination by mapping pixels. To preserve the illumination variation relations between neighboring pixels, we constrain $A$ by:

$$\mathcal{L}_A = \sum_c (t(|\nabla_x A^c|)^2 + t(|\nabla_y A^c|)^2), c \in \{R,G,B\}$$
$$t(x) = max(\alpha - |x - \alpha|, 0), \tag{14}$$

where $\nabla_x$ and $\nabla_y$ denote gradient operations along the horizontal and vertical axis, respectively, and $t$ is an identity function with a hyper-parameter $\alpha$ set to 0.02 by default.

To prevent the network from adjusting exposure solely through $d$, we additionally introduce $\mathcal{L}_B = 1 - B$ to regularize the network optimization.

In total, the overall training loss of darkening function $\mathcal{L}_D$ can be summarized as:

$$\mathcal{L}_D = \lambda_C \mathcal{L}_C + \lambda_E \mathcal{L}_E + \lambda_A \mathcal{L}_A + \lambda_B \mathcal{L}_B, \tag{15}$$

where $\lambda_C$, $\lambda_E$, $\lambda_A$ and $\lambda_B$ represent the hyper-parameters and are set to 25, 10, 1600 and 5 by default, respectively.

## C NOISE INJECTION

We follow the noise injection method of Sim-MinMax Luo et al. (2023) for the darkened images. Specifically, pixel-wise Gaussian noise $z_1$ and patch-wise Gaussian noise $z_2$ are both injected into the exposure factor $e$:

$$e = e + z_1 + z_2,$$
$$z_1 \in \mathbb{R}^{h \times w} \sim \mathcal{N}(0, \alpha_1),$$
$$\bar{z}_2 \in \mathbb{R}^{\frac{h}{d} \times \frac{w}{d}} \sim \mathcal{N}(0, \alpha_2), \tag{16}$$
$$z_2 = \text{interpolate}(\bar{z}_2, h, w),$$

where $h, w$ is the height and width of the image, $d$ is the downsampling scale, $\alpha_1, \alpha_2$ are the noise intensity both defaulting to 0.025.

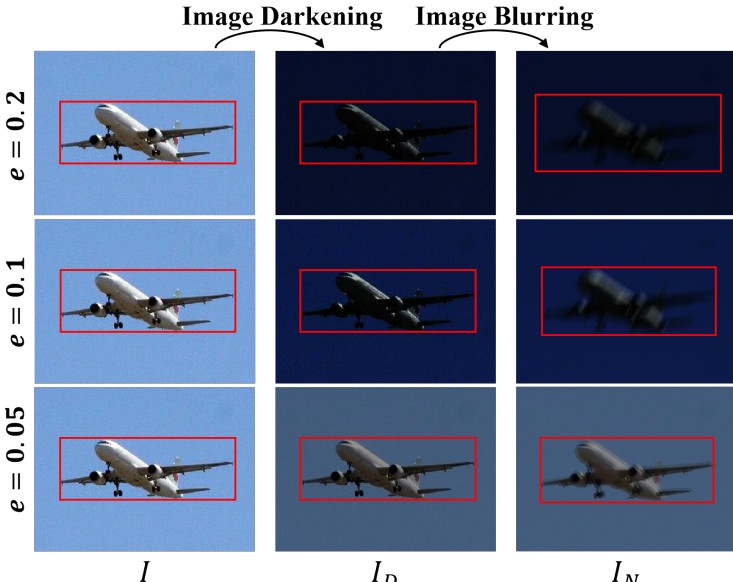

Figure 9: The synthesizing process of the ENP with different values of exposure factor $e$. The changing process of bounding boxes is also provided.

Table 5: Comparisons of different low-light synthesis methods on NightVision.

| Method | mAP |
|---|---|
| Baseline | 0.474 |
| Brightness adjustment[1] | 0.483 |
| Gamma correction | 0.490 |
| Cui *et al.* Cui et al. (2021) | 0.463 |
| Zhou *et al.* Zhou et al. (2022) | 0.472 |
| ENP (Ours) | **0.504** |

## D  COMPARISONS OF LOW-LIGHT SYNTHESIS METHODS

The Exposure-guided Nighttime Pipeline (ENP) is designed to synthesize realistic nighttime images $I_N$ by progressively darkening and blurring the daytime images $I$ under the consistent guidance of exposure factor $e$, as shown in Fig. 9.

To compare its low-light synthesis ability with other low-light synthesis methods, we adopt different low-light synthesis methods to synthesize low-light data to train the baseline YOLOv3 Redmon & Farhadi (2018). The results in Table 5 show that the ENP enhances the baseline by 3.0% mAP and outperforms other synthesis methods, verifying the effectiveness of our ENP in simulating more realistic nighttime conditions.

## E  MOTION BLUR-INDUCED FEATURE MISALIGNMENT

To quantify the feature misalignment caused by motion blur, we evaluate the mean IoU between synthetic blurry/sharp objects and region-level alignment ($\mathcal{L}_{reg}$+Box Expanding) across varying blur levels (exposure times) on NightVision.

As shown in Fig. 10, the results show that increasing blur severity could reduce IoU (Fig. 10a), amplifying spatial-semantic misalignment and consequently degrading model performance. While our region-level alignment effectively mitigates this decline (including on small objects $AP_S$) by

---

[1]https://pillow.readthedocs.io/en/stable/reference/ImageEnhance.html

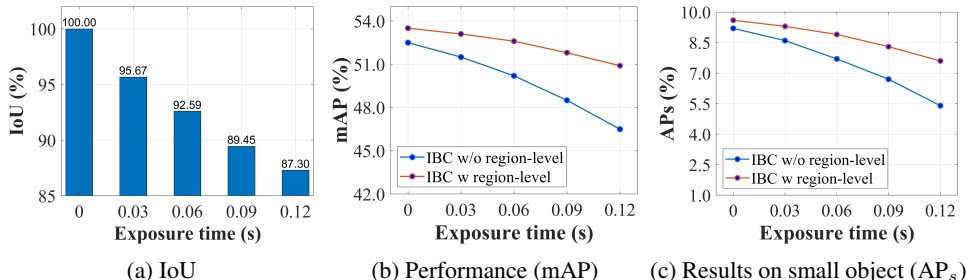

Figure 10: Study of Motion blur-induced Feature Misalignment on NightVision.

Table 6: Daytime-nighttime dataset pairs. The daytime datasets are sourced from COCO Lin et al. (2014) and PASCAL VOC Everingham et al. (2010) (VOC for short).

| Nighttime | Daytime | #Images | #Category | Source |
|---|---|---|---|---|
| NightVision | COVO$_{18}$ | 96,006 | 18 | COCO, VOC |
| ExDark | COCO$_{12}$ | 90,150 | 12 | COCO |

Table 7: Comparison results (mAP) of different methods on RAW-NOD and BDD100K.

| Method | Publication | BDD100K | RAW-NOD |
|---|---|---|---|
| YOLOv3 Redmon & Farhadi (2018) | - | 0.298 | 0.425 |
| +Retinexformer Cai et al. (2023) | *ICCV 2023* | 0.280 | 0.424 |
| +Zero-DCE Guo et al. (2020) | *CVPR 2020* | 0.299 | 0.409 |
| +LEDNet Zhou et al. (2022) | *ECCV 2022* | 0.275 | 0.410 |
| +FourierDiff Lv et al. (2024) | *CVPR 2024* | 0.269 | 0.400 |
| +DarkIR Feijoo et al. (2025) | *CVPR 2025* | 0.289 | 0.420 |
| MAET Cui et al. (2021) | *ICCV 2021* | 0.312 | 0.429 |
| CIConv Lengyel et al. (2021) | *ICCV 2021* | 0.320 | 0.429 |
| Sim-MinMax Luo et al. (2023) | *ICCV 2023* | 0.346 | 0.433 |
| DAI-Net Du et al. (2024) | *CVPR 2024* | 0.336 | 0.445 |
| **IBC (Ours)** | - | **0.386** | **0.465** |

spatially aligning features between blurry/sharp objects, demonstrating the feature misalignment and the effectiveness of our region-level alignment.

# F    MORE IMPLEMENTATION DETAILS

We train all the zero-shot models for 80 epochs on the training sets of daytime datasets and fine-tune them for 12 epochs on the union of training and validation sets of nighttime datasets. The batch size is 16. We adopt an SGD optimizer Ruder (2016), set the learning rate to 1e-3, and adopt a Cosine Learning Rate Deacy. We perform all the experiments using two Nvidia GeForce RTX 4090 GPUs on a Linux system. The results are averaged over three runs.

**Daytime-Nighttime dataset pairs.** We conduct two daytime-nighttime dataset pairs: COVO$_{18}$−NightVision and COCO$_{12}$−ExDark Loh & Chan (2019), where each subscript indicates the same number of categories as the corresponding nighttime dataset. Notably, COCO$_{12}$ is totally sourced from COCO Lin et al. (2014) while COVO$_{18}$ is additionally sampled from PASCAL VOC Everingham et al. (2010). The information about datasets is summarized in Table 6.

# G    ZERO-SHOT EVALUATIONS ON OTHER DATASETS

To further validate the effectiveness of our IBC approach, we conduct zero-shot evaluations on two additional datasets: BDD100K Yu et al. (2020) and RAW-NOD Morawski et al. (2022). For BDD100K, we partition the dataset following its original annotations, utilizing daytime scenes for training and nighttime scenes for evaluation. For RAW-NOD, we construct a training set by sam-

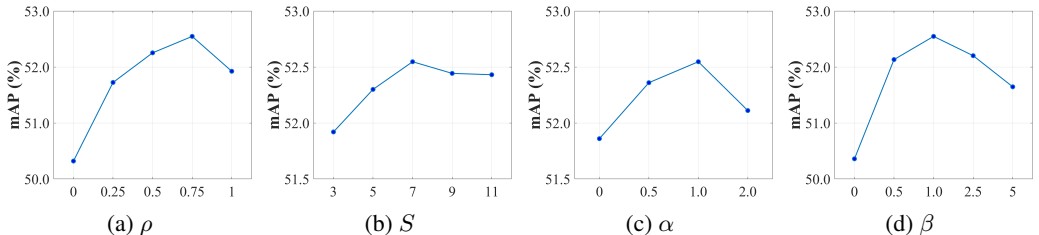

Figure 11: Analysis of different hyper-parameters on NightVision.

Table 8: Analysis of exposure factor $e$ on NightVision.

| Dataset | [0, 0.1] | | [0, 0.15] | | [0, 0.2] | | [0, 0.25] | |
|---|---|---|---|---|---|---|---|---|
| | Separate | Joint | Separate | Joint | Separate | Joint | Separate | Joint |
| NightVision | 0.508 | 0.519 (+1.1%) | 0.509 | 0.522 (+1.3%) | 0.510 | **0.525 (+1.5%)** | 0.509 | 0.522 (+1.3%) |
| ExDark | 0.521 | 0.530 (+0.9%) | 0.523 | 0.534 (+1.1%) | 0.523 | **0.539 (+1.6%)** | 0.523 | 0.536 (+1.3%) |

Table 9: Ablation results (mAP) of Box Expanding.

| Dataset | IBC (Ours) | w/o Box Expanding |
|---|---|---|
| **NightVision** | **0.525** | 0.506 (-1.9%) |
| **ExDark** | **0.539** | 0.521 (-1.8%) |

pling daytime images from COCO Lin et al. (2014) according to RAW-NOD's object categories, while employing the RAW-NOD test set for evaluation. All experimental configurations maintain consistency with those established in the main text. The results (mAP) are provided in Table 7.

As shown in Table 7, our method consistently outperforms other competitive enhancement and zero-shot methods on both BDD100K and RAW-NOD datasets. It can be observed that due to the consideration of motion blur in low-light environments, our method achieves a higher performance improvement (8.8% mAP) over the baseline on BDD100K dataset which contains a large number of traffic dynamic scenarios. These results further demonstrate the superior zero-shot generalizability of our method in low-light object detection.

## H  MORE ABLATION STUDIES

### H.1  HYPER-PARAMETERS ANALYSIS

**The probability $\rho$ of image blurring.** Fig. 11a examines the impact of $\rho$. The results indicate that the involvement of motion blur brings general enhancements where $\rho = 0.75$ achieves the best result.

**The size $S$ of region features.** Fig. 11b ablates the impact of $S$. The results indicate that performance rises initially and then falls as S increases. One possible explanation is that a small $S$ loses spatial information of large-sized objects, while a large $S$ results in sparse spatial information of small-sized objects, and $S = 7$ achieves the best balance.

**The balanced factor $\alpha$.** Fig. 11c examines the impact of $\alpha$ in Eq.11. It can be observed that the involvement of the instance-level consistency loss contributes to enhancing the performance, and $\alpha = 1.0$ yields the best result.

**The balanced factor $\beta$.** Fig. 11d ablates the impact of $\beta$ in Eq.11. The results show that when $\beta = 1.0$, the multi-level consistency loss could regularize the detector to achieve the best performance.

**The exposure factor $e$.** As shown in Table 8, we analyze separate (different $e$) and joint (shared $e$) controls of brightness/blur within different ranges, where the results (mAP) justify our joint control's superiority, with [0, 0.2] yielding the best.

Figure 12: Visualization of Synthetic Blurry Evaluation.

Table 10: Comparison results (mAP) on RAW nighttime data.

| Methods | LIS (RAW) | | |
|---|---|---|---|
| | Original | Syn. Blurry | $\Delta$ (%) |
| Faster-RCNN | 0.374 | 0.253 | -12.1 |
| Chen et al. (2023) | 0.566 | 0.453 | -11.3 |
| IBC (Ours) | **0.587** | **0.531** | **-5.6** |

Table 11: Comparison results (mAP) on daytime data.

| Datasets | | YOLOv3 | CIConv | MAET | Sim-MinMax | DAI-Net | IBC (Ours) |
|---|---|---|---|---|---|---|---|
| COVO$_{18}$ | Ori. | 0.505 | 0.506 | 0.511 | 0.516 | 0.513 | **0.521** |
| | Syn. | 0.417 | 0.415 | 0.420 | 0.425 | 0.420 | **0.465** |
| COCO$_{12}$ | Ori. | 0.526 | 0.529 | 0.532 | 0.533 | 0.530 | **0.541** |
| | Syn. | 0.426 | 0.419 | 0.411 | 0.430 | 0.427 | **0.483** |

## H.2 BOX EXPANDING

To study the effectiveness of Box Expanding, we remove this operation from our IBC. The results in Table 9 show that removing Box Expanding greatly degrades the performance of IBC, due to the feature misalignment between blurred/original objects. This also verifies the effectiveness of Box Expanding.

## I VISUALIZATION OF SYNTHETIC BLURRY EVALUATION

Here we provide the visualization of Synthetic Blurry Evaluation to highlight the effectiveness of our method. Examples in Fig. 12 show that our method can accurately localize both sharp and synthetic blurry objects while others fail.

## J GENERALIZATION TO OTHER DOMAINS

**RAW Nighttime Domain.** RAW-based low-light detection is a compelling direction since RAW images contain more original information than sRGB data Chen et al. (2023); Guo et al. (2025). To study the generalizability of IBC over RAW images, we replace our darkening pipeline with the RAW-based synthetic pipeline of Chen et al. (2023), and motion blur is generated on the demosaicked RAW images. The models are trained on the COCO Lin et al. (2014) and evaluated on the LIS dataset Chen et al. (2023). Notably, we did not conduct real-world blurry evaluation as LIS is collected under a stable system, hence there were no blurry images. The baseline is Faster-RCNN Ren et al. (2016). The results in Table 10 justify our method which significantly outperforms the baseline and presents less performance degradation in synthetic blurry RAW images.

**Daytime Domain.** In addition to low-light data, we also evaluate the methods on daytime datasets in Table 11. These results verify that our method remains generalizable over original (Ori.) and synthetic blurry (Syn.) daytime data.

## K ETHICS STATEMENT

The images of NightVision are CC0/CC BY 4.0 licensed, and all image provenance is included in the metadata to avoid potential copyright disputes. Before the release of NightVision, all identifiable content will undergo verification and blurring to safeguard privacy. The dataset is for academic purposes only and not for commercial usage. We confirm that we bear all responsibility in case of violation of rights during the collection of data on NightVision, ensuring accountability and commitment to maintaining ethical standards. We will take appropriate action when needed.

## L THE USE OF LARGE LANGUAGE MODELS (LLMs)

In this work, we only used large language models to correct grammar errors and polish the writing.

