# OpenReview forum: "Improving Zero-shot Low-light Object Detection via Handling of Motion Blur"
_ICLR.cc/2026/Conference — ICLR 2026 Conference Withdrawn Submission_

### Official Review · Reviewer_tpMS · 2025-10-27

**Soundness:** 3
**Presentation:** 3
**Contribution:** 2
**Rating:** 4
**Confidence:** 5

**Summary:**

This paper proposes an Illumination-Blur Consistency framework for zero-shot low-light object detection. It builds upon prior zero-shot day-night adaptation methods (especially Similarity Min-Max), which modeled illumination consistency through controllable exposure-guided image darkening and feature-level similarity maximization. The new contribution introduces motion blur as an additional degradation factor co-occurring with low illumination. They also release a new low-light dataset, NightVision, containing 10K images and 18 categories. Experiments on ExDark and NightVision report moderate gains over previous zero-shot baselines including Sim-MinMax.

**Strengths:**

- **Writing.** The paper is clearly written and easy to understand

- **Clear motivation.** The paper presents a natural extension of the illumination-consistency idea from Sim-MinMax to include motion blur. The proposed exposure-guided blur generation uses simple but physically interpretable camera-motion trajectories.

- **Extensive experiments.** Experiments cover both synthetic and real blurry data and include ablation studies confirming that the blur component provides measurable improvements.

**Weaknesses:**

- **Incremental novelty.**
The IBC framework reuses the core exposure-guided darkening function and similarity-based consistency training introduced in Sim-MinMax. The main new element is the addition of a blur kernel generator controlled by the same exposure factor and a region-level feature alignment. Conceptually, this is a straightforward extension of the existing framework rather than a fundamentally new idea.

- **Limited methodological depth.** The multi-level consistency loss is essentially a variant of existing contrastive consistency learning (BYOL), also used in Sim-MinMax without deeper theoretical insight.

- **Lack of broader insight.**
The paper still focuses narrowly on illumination and blur, with little discussion of other nighttime factors (e.g., sensor noise, non-uniform lighting, glare). The “illumination-blur equivariance” claim remains qualitative.

Overall I think this paper looks like an extension of the Sim-MinMax paper rather than a introducing a brand-novel approach towards the ZLOD task. Although some components are proposed by this paper, e.g., the blur synthesis module, I think this is not enough to champion the acceptance of this paper. Nevertheless, I acknowledge this paper proposes a valid approach and achieves SoTA results. Therefore, I recommend borderline reject.

**Questions:**

- Have the authors considered joint training and the blurring function and the detection network or using learned blur kernels? Currently the deblurring module is completely based on heuristics. How sensitive are results to the manually designed camera-motion trajectories?

- It would strengthen the paper to show the darkening results of Sim-MinMax and the proposed method.

---

### Official Review · Reviewer_psuM · 2025-10-31

**Soundness:** 2
**Presentation:** 2
**Contribution:** 2
**Rating:** 2
**Confidence:** 5

**Summary:**

To address the low light and motion blur, the paper provides the NightVision dataset and proposes an Illumination-Blur Consistency (IBC) framework for Zero-shot low-light object detection (ZLOD). In detail, IBC first synthesizes nighttime images with image darkening and motion blur generation. And then the paper explores illumination-blur equivariant representations at the region and instance levels. Experimental results demonstrate the effectiveness of the proposed method on the proposed NightVision dataset and the ExDark dataset.

**Strengths:**

The methodology section is presented in a formal, equation-based manner, and figures are aesthetically rendered.

**Weaknesses:**

1.	In the "motion blur learning" section of the related work, how do existing methods address motion blur, particularly in the field of object detection? There is a lack of systematic summarization. Furthermore, the distinction between the proposed method in this paper and existing approaches for handling motion blur has not been clarified.
2.	In the "Low-light Object Detection" section of the related work, only a brief description is provided regarding how zero-shot object detection methods operate, while the intrinsic connections and differences among them are not discussed.
3.	In Lines 277–278, why select the last three feature layers to ensure consistency at both the region and instance levels?
4.	The Exdark dataset considers low illumination, motion blur, multi-angle, multi-category, and multi-weather conditions. Compared to Exdark, what is the core advantage of the dataset proposed in this paper?
5.	The methodological innovation is limited. Compared to existing zero-shot low-light object detection methods, this paper merely extends the consideration to motion blur. As shown in Figure 1, the overall pipeline remains unchanged from previous approaches.
6.	Currently, there are integrated algorithms proposed for low-light image enhancement and deblurring, such as [1]. What would be their performance if applied to the zero-shot low-light object detection domain?

[1] Lv X, Zhang S, Wang C, et al. Fourier priors-guided diffusion for zero-shot joint low-light enhancement and deblurring[C]//Proceedings of the IEEE/CVF Conference on Computer Vision and Pattern Recognition. 2024: 25378-25388.

**Questions:**

1.	What is the innovation of the proposed method in this paper compared to existing motion blur learning approaches?
2.	Compared to existing zero-shot low-light object detection methods, apart from further considering motion blur, what is the innovation of the proposed method in this paper?
3.	The Exdark dataset considers low illumination, motion blur, multi-angle, multi-category, and multi-weather conditions. Compared to Exdark, what is the core advantage of the dataset proposed in this paper?
4.	In Lines 277–278, what is the rationale for selecting the last three feature layers to ensure consistency at both the region and instance levels?
5.	Currently, there are integrated algorithms proposed for low-light image enhancement and deblurring, such as [1]. What would be the effect if they were applied to the zero-shot low-light object detection domain?

[1] Lv X, Zhang S, Wang C, et al. Fourier priors-guided diffusion for zero-shot joint low-light enhancement and deblurring[C]//Proceedings of the IEEE/CVF Conference on Computer Vision and Pattern Recognition. 2024: 25378-25388.

---

### Official Review · Reviewer_nbLy · 2025-11-08

**Soundness:** 3
**Presentation:** 3
**Contribution:** 3
**Rating:** 6
**Confidence:** 4

**Summary:**

This paper aims to address the core challenge of ZLOD, namely domain generalization without target dark data. It points out that existing methods only focus on illumination consistency while neglecting the motion blur associated with long camera exposure times. To tackle this issue, the paper proposes the IBC framework: the ENP regulates illumination reduction and motion blur generation via a single exposure factor to synthesize realistic nighttime images; the MMA resolves feature misalignment through region-level and instance-level losses to learn illumination-blur equivariant representations. Meanwhile, the paper constructs the NightVision dataset, covering multi-angle, multi-weather, and multi-size scenarios, with diversity surpassing existing benchmarks. Experimental results demonstrate that IBC outperforms zero-shot and enhancement-based methods on multiple datasets.

**Strengths:**

This paper accurately captures the coexistence of motion blur and low-light conditions in ZLOD. The proposed IBC framework achieves efficient domain generalization: The ENP regulates light reduction and motion blur generation. Formula 3 incorporates comprehensive velocity information to model positions, while the exposure factor e can regulate exposure time—both lay the foundation for generating realistic datasets. To address the mismatch between the bounding boxes of blurred objects and static objects, the MMA performs feature alignment at two scales (region-level and instance-level), resolving the issue of feature misalignment. Additionally, the authors present a comprehensive NightVision dataset. The experiments are rich in content and have a clear structure.

**Weaknesses:**

Is it reasonable to use a single exposure factor to uniformly regulate illumination and blur in ENP? It ignores practical interfering factors such as camera image stabilization and object motion speed, Specifically, the interference of the camera's anti shake mechanism and the motion state of objects in real scenes were not taken into account.; the ablation experiments fail to eliminate the interference of module interaction, and the contribution of MMA requires experimental clarification. Reference formats are inconsistent and need to be standardized.

**Questions:**

1. Regarding the design of "uniformly regulating illumination and blur with a single exposure factor" in ENP, could you supplement validation experiments in real-world scenarios to further support its physical rationality?

2. Detection methods specifically optimized for motion blur have not been included in the comparative experiments. Could you supplement the relevant comparison results?

3. The existing ablation experiments fail to eliminate the interaction interference between ENP and MMA, and it is necessary to clarify the unique contribution of MMA in addressing feature misalignment.

4. Is it possible to achieve more accurate modeling of motion trajectories based on Kalman filtering, which could potentially enhance the accuracy and robustness of trajectory estimation by effectively accounting for noise and dynamic variations in real-world shooting scenarios?

---

### Note · Authors · 2025-11-12

I have read and agree with the venue's withdrawal policy on behalf of myself and my co-authors.